OPA1 supports mitochondrial dynamics and immune evasion to CD8+ T cell in lung adenocarcinoma

Wang Ying 1
Li Yadong 2
Jiang Xuanwei 3
Gu Yayun 3
Zheng Hui 1
Wang Xiaoxuan 4
Zhang Haotian 5
Wu Jixiang ycsywjx@163.com 6 7
Cheng Yang chengyangjsnj@163.com 1
1 Center for Health Management, Jiangsu Province Geriatric Hospital , Nanjing , China
2 Department of Thoracic Surgery, The Second Clinical Medical College of Nanjing Medical University , Nanjing , China
3 State Key Laboratory of Reproductive Medicine, Center for Global Health, School of Public Health, Nanjing Medical University , Nanjing , China
4 State Key Laboratory of Translational Medicine and Innovative Drug Development, Jiangsu Simcere Diagnostics Co., Ltd. , Nanjing , China
5 The First Clinical Medical College of Nanjing Medical University , Nanjing , China
6 Department of Thoracic and Cardiovascular Surgery, The Yancheng School of Clinical Medicine of Nanjing Medical University , Yancheng , China
7 Department of Thoracic and Cardiovascular Surgery, The Sixth Affiliated Hospital of Nantong University , Yancheng , China
Yan Yuanliang
Electronic publication date: 2022 Dec 21
Publication date: 2022
Volume: 10
Electronic Location ID: e14543
Received 2022 Aug 9; Accepted 2022 Nov 18
Copyright: ©2022 Wang et al.
Copyright year: 2022
Copyright holder: Wang et al.
License: This is an open access article distributed under the terms of the Creative Commons Attribution License, which permits unrestricted use, distribution, reproduction and adaptation in any medium and for any purpose provided that it is properly attributed. For attribution, the original author(s), title, publication source (PeerJ) and either DOI or URL of the article must be cited.
License URL: https://creativecommons.org/licenses/by/4.0/

Keywords: OPA1, Lung adenocarcinoma, Mitochondrial fusion, Immune evasion, CD8+ T cell

Funding: National Natural Science Foundation of China 81903385 and 81902836 This work was supported by grants 81903385 and 81902836 from the National Natural Science Foundation of China. The funders had no role in study design, data collection and analysis, decision to publish, or preparation of the manuscript.

==============================
Background

Mitochondrial fusion and fission were identified to play key roles during multiple biology process. Thus, we aim to investigate the roles of OPA1 in mitochondria fusion and immune evasion of non-small cell lung cancer cells.

Methods

The transcriptional activation of genes related to mitochondrial dynamics was determined by using multi-omics data in lung adenocarcinoma (LUAD). We elucidated the molecular mechanism and roles of OPA1 promoting lung cancer through single-cell sequencing and molecular biological experiments.

Results

Here, we found that copy number amplification of OPA1 and MFN1 were co-occurring and synergistically activated in tumor epithelial cells in lung cancer tissues. Both of OPA1 and MFN1 were highly expressed in LUAD tumor tissues and OPA1 high expression was associated with poor prognosis. In terms of mechanism, the damaged mitochondria activated the apoptotic signaling pathways, inducing cell cycle arrest and cell apoptosis. More interestingly, OPA1 deficiency damaged mitochondrial dynamics and further blocked the respiratory function to increase the sensitivity of tumor epithelial to CD8+ T cells in non-small cell lung cancer.

Conclusions

Our study demonstrated the high co-occurrence of copy number amplification and co-expression of OPA1 and MFN1 in LUAD tissue, and further revealed the contribution of OPA1 in maintaining the mitochondria respiratory function and the ability of immune evasion to CD8+ T cells of LUAD.

Introduction

Lung cancer is the leading deadly malignancy worldwide (Bray et al., 2018), among which non-small cell lung carcinoma (NSCLC) accounts for more than 85% of cases (Wang et al., 2019). Patients diagnosed with NSCLC have an overall 5-year survival rate of less than 18% (Zappa & Mousa, 2016). Thus, a deeper exploring of the underlying mechanisms in NSCLC cell progression is crucial for developing effective treatments.

Mitochondrial dynamics include reshaping, rebuilding, and recycling events of mitochondria (Sharma et al., 2021), controlled by a growing family of “mitochondria-shaping” proteins (Griparic & van der Bliek, 2001). Dynamin 1 like (DNM1L) and fission 1 (Fis1) (Forrester et al., 2020) are required for fission process. During fission DNM1L translocates to outer mitochondrial membrane (OMM) where it interacts with Fis1 to complete organelle division (Frezza et al., 2006). Mitofusins (MFN1/2) (Eura et al., 2003; Santel et al., 2003) and optic atrophy 1 (OPA1) (Ehses et al., 2009) mediate mitochondrial fusions of outer and inner mitochondrial membrane (IMM) respectively.

A growing body of evidence suggests that mitochondrial fusion and fission participate in various cellular activities, including oxidative stress (Yi et al., 2019; Yu et al., 2017; Zhang et al., 2018), apoptosis (Morita et al., 2017; Pena-Blanco & Garcia-Saez, 2018; van der Bliek, Shen & Kawajiri, 2013), mitophagy and so on (Chen et al., 2016; van der Bliek, Shen & Kawajiri, 2013; Yoo & Jung, 2018), which are closely related to tumorigenesis and tumor progression (Simula, Nazio & Campello, 2017). Among dynamin-related GTPases mentioned above, OPA1, as the fusion protein, was a bifunctional protein. On one hand it promotes mitochondrial fusion, depending on MFN1. On the other hand, it regulates apoptosis by controlling cristae remodeling and cytochrome c redistribution, and this occurs independently from mitochondrial fusion. Thus, we systematically investigated the mechanism of the related gene expression activation, and especially the potential role of OPA1 in NSCLC.

Besides, the inflammatory cell infiltrates formed in human cancers could promote natural disease progression or, conversely, contribute to antitumor effects (van der Leun, Thommen & Schumacher, 2020). It is well documented that CD8+ T cells have the ability to recognize and eliminate cancer cells (van der Leun, Thommen & Schumacher, 2020). Here, we found that copy number amplification of OPA1 and MFN1 were co-occurring and synergistically activated in lung cancer tissues. Moreover, activation of OPA1 promoted mitochondrial dynamics in tumor epithelial cells to escape CD8+ T cells killing.

Materials and Methods

Acquisition and processing of TCGA database

Transcriptome RNA sequencing (RNA-seq) data of 1,111 NSCLC samples (LUAD: 513 tumor tissues and 59 matched adjacent tissues; lung squamous cell carcinoma, LUSC: 489 tumor tissues and 50 matched adjacent tissues) were obtained from the TCGA database (https://portal.gdc.cancer.gov/) with level 3. The association between OPA1 and MFN1 expression and survival in all tumor tissue samples was analyzed by Cox regression, which was obtained from http://www.oncolnc.org/cancer/.

Immunohistochemical analysis (IHC)

All of the tissues were handled with the following steps: (1) Deparaffinizing and rehydrating the paraffin section; (2) Antigen retrieval; (3) For cooling to room temperature before proceeding, the sections were placed in PBS (pH = 7.4) and shaken on a decolorization shaker 3 times for 5 min each; (4) Blocking endogenous peroxidase activity; (5) Primary antibody incubation and secondary antibody incubation; (6) DAB chromogenic reaction nuclear counterstaining, dehydration and mounting. Finally, staining of the tissues were visualized with a microscope, and images were acquired and analyzed. H-score: The depth and quantity of positivity was scored by Quant Center -an analysis software matched with a 3D scanner, which can only quantify the brown–yellow color of DAB. The larger the value, the stronger is the positivity. The assay refers to the methods in previously reported research (Yin et al., 2021).

ATP Assays

ATP was analized by enhanced ATP assay kit (Beyotime, S0027). The assay was performed ref to the protocols reported by previous studies (Kuang et al., 2021).

Protein isolation, Western blotting and antibodies

Proteins were extracted from cells with RIPA and protease inhibitor: phenylmethanesulfonyl fluoride (Beyotime, Haimen, China) and cocktail (MedChemExpress, Shanghai, China). The BCA method was used to measure protein concentrations. Western blotting was performed as previously described (Xie et al., 2018). The antibodies used were as follows: anti-OPA1 (1:1000, ab157457; Abcam); anti-cytochrome C (1:5000, ab133504; Abcam); and anti-GAPDH (1:1000, AG019; Beyotime).

Cell culture and transfection

Human LUAD cell lines (NCI-H1299 and NCI-A549) were bought from the Shanghai Institute of Biochemistry and Cell Biology. Cells were cultured in DMEM (Gibco, Carlsbad, MA) supplemented with 10% FBS (Gibco). These cells were incubated at 37 °C with 5% CO2 in a humidified incubator.

The shRNAs specific for OPA1 were synthesized (RiboBio, Guangzhou, China; target sequence 1#: cgGGAGTTTGATCTTACCAAA and target sequence 2#: ccGGACCTTAGTGAATATAAA), and the overexpression plasmid targeting OPA1 was custom-designed (GENOME, Nanjing, China). The plasmid DNA or shRNA was transiently transfected into cells with Lipofectamine 2000 reagent (Invitrogen, Shanghai, China).

Cell proliferation assay

Cell proliferation was analyzed by CCK8 (Dojindo, Japan) and colony formation assays per the manufacturer’s instructions. A total of 4 ×103 cells were seeded in 96-well plates (Corning Incorporated, Corning, New York, USA), and the culture medium was replaced with 10 µl CCK8 solution mixed with 100 µl RPMI 1640 or DMEM every 1 day and then incubated at 37 °C. The optical density value was measured using a Multimode microplate reader and obtained from five technical replicates (450 nm; Tecan, Mechelen, Belgium). 1 ×103 cells were seeded in 6-well per plate (Corning Incorporated, Corning, New York, USA) and maintained for 14 days. Cell colony formation abilities were fixed with methanol and then stained with crystal violet (Beyotime, Haimen, China) for 30 min. All wells were photographed and counted. Each assay was performed at least three times. To evaluate the effects of mitochondrial target treatment on cell proliferation, cells were treated with N-(1,5-dimethyl-3-oxo-2-phenyl-2,3-dihydro-1H-pyrazol-4-YL)-3 methyl-1-PH+ (MYLS22, 50 µM) or Oligomycin A (10 µM) when the complete DMEM medium was added. Then, CCK8 and colony formation assays were performed to estimate the cell proliferation ratio of different treated cells.

Transwell assays

The migration and invasion capacity of lung cancer cells were investigated using Transwell plates (Corning Incorporated, Corning, NY, USA). Culture medium containing 10% FBS was added to the bottom compartment of the chamber as a chemoattractant. A total of 2 × 104 cells/100 µl of lung cancer cells were planted on the upper chamber in serum free solution. Cells that adhered to the transwells were fixed with methanol for 15 min and then stained with crystal violet (Beyotime, Haimen, China), and quantified from five averaged fields via a Q-fired cooled CCD camera attached to an Olympus microscope. The assay was repeated three times in duplicate. To evaluate the effects of mitochondrial target treatment on cell migration, cells were treated with MYLS22 (50 µM) or Oligomycin A (10 µM) when migration was induced. Then, transwell based assays were performed to estimate the cell migration ability of different treated cells.

Mitochondrial membrane potential (Δψm)

Cells of different groups were incubated in tetramethylrhodamine methyl ester (TMRM; Thermo her Scientific, T-668) (50 nM). We obtained the images with LSM800 confocal microscope, equipped with a 60  × 1.40 NA oil immersion objective. Then the cell fluorescence intensity was analyzed with software ImageJ. Mitochondria with normal cells fluoresce bright red staining; however, red staining becomes diffuse or very light when mitochondrial membrane potential (Δψm) dissipates.

Mitochondrial Oxygen consumption rate (OCR) determination

Mitochondria oxygen consumption rate (OCR) was analyzed by using the Seahorse Extracellular Flux (XF96) Analyzer in group of both sh-OPA1 and control cells (Seahorse Bioscience Inc, USA). NCI-H1299 cells were plated at 4 × 105 cells per well in the Seahorse V7 microplate for 24 h in the incubator. Then, the above cells were incubated in XF medium without NaHCO3 and FBS for 1 h/37 °C without CO2 input. OCR was calculated as percentage of the OCR value before the treatment of tested agents (pMoles/minute). The respiratory capacity was analyzed after the treatment with oligomycin and FCCP. OCR reduction after antimycin A, an inhibitor of ATP synthase, treatment represents ATP turnover under specific condition.

Metabolomics analysis

1 × 107 cells in different groups (group of Control and shRNA-OPA1, n = 3/group) were seeded in 10 cm culture dishes. Then the cells were harvested and kept in liquid nitrogen. The metabolites were extracted and subjected to analysis of central carbon metabolism (Zhang et al., 2021).

Analysis of cell cycle progression and apoptosis using flow cytometry

Cells were harvested using 0.25% trypsin and washed with PBS. After resuspending in PBS and counting, 2 ×105 cells were isolated and centrifuged to obtain a cell precipitate. Annexin V-FITC and propidium iodide (PI) staining solution were added with low-speed shaking and keep in the dark for 10 min.

Cells were re-suspended in 70% cold ethanol and 4 ° C overnight then centrifuged and washed in PBS. 150 µl RNase A and PI staining solution were then added to each tube and were then incubated at 4 °C. 20 min later, the results were analyzed by using flow cytometry and the ratio of cell among the G1, S and G2/M phases were analyzed (Zuo et al., 2014).

Quantitative real-time RT-PCR (qRT-PCR)

The cDNA obtained by reverse transcription was amplified on a QuantStudio™ 7 Flex Real-Time PCR System. The expressions of target genes were determined using the 2−ΔCt method relative to GAPDH. The sequence of primers are presented in Table S1.

RNA sequencing and analysis

OPA1 knockdown and control LUAD cells were conducted using RNA-seq as described previously (Wang et al., 2018). The mRNA was extracted from total RNA using oligo (dT) beads to obtain the mRNA-seq library according to a standard protocol (Illumina, San Diego, USA) and then sequenced using an Illumina HiSeq X Ten system. The statistical power of this experimental design, was calculated in Source Package RNASeqPower in R (version 4.2) with the calculated power of 0.89. Three biological replicates were involved in both of the two groups for the RNA-seq analysis. 6G, 294 × depth raw data per sample were filtered and processed by Q30 and aligned to the mouse reference genome GRCh37/Gencode v19 with STAR software (v2.5.3a). Then, assembly and quantification of the transcripts were accomplished with RESM software (Version 1.3.3) guided by the Ensembl gtf gene annotation file (http://www.ensembl.org/). Read counts were used for the measurements of the relative abundance of the transcripts. The differential expressed genes (DEG) between groups were identified by the Wilcoxon rank sum test (Liu et al., 2021) (P < 0.05 and —log2[fold change]—>1). DAVID was further used to conduct an enrichment analysis of the DEGs (version 6.8; https://david.ncifcrf.gov/). The top 5 GO terms of the biological processes were examined. GSEA was performed using the R package cluster Profiler as previously reported (Chen et al., 2020).

Single-cell RNA sequencing (scRNA-seq) statistical analysis

The 10× Genomics scRNA-seq matrix of early stage LUAD tumor from primary lung tissues (tLUNG) were obtained from the previous study (Kim et al., 2020). The filtered and batched digital gene expression matrix (UMI counts per gene per cell) of 11 LUAD tLUNG samples was imported in R version 4.1 using Seurat v4.1.0. In our study, we used the t-Distributed Stochastic Neighbor Embedding (tSNE) algorithm for visual processing of this dimension-reduced dataset. Besides, main cell types were identified by scoring canonical cell type markers across clusters (Bischoff et al., 2021).

After identifying cell types in LUAD tLUNG samples, cell–cell communication was analyzed by implementing the Cell Chat R package v1.4.0 and used the Cell Chat website https://github.com/sqjin/CellChat. A new CellChat object was created from the merged Seurat object. The dataset of human CellChatDB was set as referencing database. Next, the communication probability was computed using the function computeCommunProb with the default parameters. After that, the cell–cell communication was inferred and the cell–cell communication network was aggregated with default parameters. The number of interactions was visualized to show the aggregated cell–cell communication network and signaling sent from each cell cluster.

Statistical analysis

We provided our raw numeric data for review and publication so the statistical analysis performed in our study can be checked (as shown in Tables S2–S7). All statistical analyses were performed by using R software. For experimental data, the Student’s t test or one-way ANOVA was used to assess differences in treatment groups using GraphPad Prism software (version 6.01). The results are expressed as the mean ± s.e.m. of three independent experiments. The significance threshold was set as a P value <0.05.

Results

Identification of the clinical and prognostic value of mitochondrial dynamics-related genes in NSCLC from TCGA by bioinformatic analysis

To systematically evaluate the mechanism of mitochondria dynamics in NSCLC development, we performed the differential expression analysis of the five classical mitochondrial dynamics-related genes between the adjacent and tumor samples in the TCGA database (Fig. 1A). We identified that OPA1, MFN1, and DNM1L were significantly increased in both NSCLC tumor tissues (P < 0.05). However, Fis1 was decreased in tumors which implying that mitochondrial fusion is more active in tumors than mitochondrial fission. Fis1 is an important recruiting protein on the OMM. In prokaryotic cells, Fis1 assembles with DNM1L on the OMM to form a DNM1L ring and initiates mitochondrial fission by ring contraction and scission. Interestingly, in some cases, Fis1 is not necessary for mitochondrial fission, and the cells with Fis1 knockout can replicate and proliferate without obvious changes in mitochondrial morphology. In our study, we found that DNM1L was highly expressed in tumor tissues, while Fis1 was low expressed. Thus, we speculated that DNM1L may control fission through the regulation of other structural proteins than Fis1. To further explore the mechanism of the transcriptional activation of fusion related genes, we used a combined analysis with genomic data and found OPA1 and MFN1 have higher genomic alteration rate of 17% and 20%, with copy number variation (CNV) as the most common mutation type (Fig. 1B). The alteration type of the two genes is shown in Fig. 1C, mainly consisting of gain and amplification. Further analysis found the significant correlation of genomic amplification and gene expression of both OPA1 and MFN1 (Fig. 1D). It is worth noting that NSCLC patients with OPA1 and MFN1 gain or amplification tended to have co-occurrence implying their synergistic roles in mitochondrial dynamics. Immunohistochemical results also revealed that the protein expression of OPA1 was significantly increased in LUAD tumor tissues (Fig. 1E). In addition, Kaplan–Meier curves revealed that higher expression of OPA1 was significantly correlated with poor survival in LUAD patients, while other proteins does not (Fig. 1F). These results suggest that OPA1 plays a stronger role in tumorigenesis than other mitochondrial fusion and fission related genes, possibly because it has other functions independent of mitochondrial dynamics.

Figure 1 The clinical and multi-omics data of mitochondrial dynamics-related genes (MFN1/2, OPA1, DNM1L and Fis1) in NSCLC tumor tissues.

(A) The differently expressed of MFN1/2, OPA1, DNM1L and Fis1 in tumor and adjacent tissues based on TCGA database. (B) Schematic diagram showing somatic alterations in the genes identified in NSCLC. Amplification (red) was displayed. (C–D) Copy number alterations of OPA1 and MFN1 in NSCLC samples from TCGA data (C) and a box plot showing the association between mRNA levels and gene amplification or deletion (D). (E) OPA1 representative IHC stained images in tumor and adjacent tissues (n = 55). (F) Kaplan–Meier survival curves of OPA1and MFN1 expression in LUAD from TCGA.∗P < 0.05; ns, not significant.

Loss of OPA1 induced mitochondrial dysfunction and metabolic reprogramming

We further examined transcriptional datasets from shOPA1 and control cells, and the PCA is shown in Fig. S1A. With a filter P value <0.05 and —log2Fold-change—>1, significantly different genes consisting of 1,156 down-regulated genes and 1,470 up-regulated genes in LUAD are shown in a volcano plot (Fig. 2A). Clustering analysis is shown in the heatmap in Fig. 2B. Specifically, we performed a GO analysis of these genes and revealed that they were involved in many signaling pathways, including mitochondrial function, ATP binding, ATP-dependent DNA helicase activity and membrane pathways (Fig. 2C), which provided clues about mitochondria. KEGG & COG_ONTOLOGY analysis also revealed dysfunction of mitochondrial metabolism (Figs. S1B & S1C). Furthermore, GSEA was performed to verify that the function of mitochondria and ATP production were inhibited in ShOPA1 cells (Figs. 2D and 2E). Furthermore, we conducted electron microscopy, and the representative images were shown in Fig. 2F, we identified the significantly reduced length/width ratio of mitochondrial in shOPA1 cells compared to the cell in control groups. To confirm the effect of OPA1 deficiency on ATP levels, we measured the production of ATP in shOPA1 and control cells, and the results showed that the ATP concentration of the OPA1 deficiency group was lower than that of the control group (Fig. 2G). These data inspired us to explore more about the aberrant metabolism caused by OPA1 deficiency.

Figure 2 Loss of OPA1 induced mitochondrial dysfunction and decreased ATP production.

(A) Volcano plot of statistical significance (P < 0.05) against fold change (ratio of ShOPA1/Control group), demonstrating the most significantly differentially expressed genes by genome-wide transcriptomic analysis between Control and ShOPA1 NCI-H1299 cells. (B) Heat map of color-coded expression levels of differentially expressed genes (two-way ANOVA (n = 3)). (C) GO pathway enrichment analysis within the complete set of differentially expressed genes. (D, E) GSEA enrichment plots showing that loss of OPA1 results in the dysfunction of MITOCHONDRIA_GENE_MODULE and ATP_METABOLIC_PROCESS gene set. (F) Transmission electron microscopy was used to observe mitochondrial morphology and combined with Image J measurement to analyze the changes in the area and number in Control and ShOPA1 NCI-H1299 cells. Scale bars, 500 nm. (G) ATP production was detected in Control and ShOPA1 cells. Abbreviations: CC, Cellular Component; BP, Biological Process; MF, Molecular Function. For A to G, n = 3 and the bars represent mean ±SEM. The statistical analysis was carried out using t-test, NS denotes no significant, ** denotes P < 0.01.

OPA1 modulates the TCA metabolic process in LUAD cells

The mitochondrial metabolism of central carbon metabolism and oxygen consumption were further measured in NCI-H1299 cell lines. The obtained score plot of PCA analysis enabled a clear separation between the control group and the shOPA1 group (Fig. 3A). In addition, we identified 49 differentially abundant metabolites, consisting of 30 down-regulated and 19 up-regulated metabolites, based on a volcano plot (Fig. S1D). Variable importance in projection showed that significantly different metabolites were mainly involved in the processes of TCA and glycolysis (Fig. 3B). The relationship between the samples and metabolites is shown in Fig. S1E. When these significantly altered metabolites were subjected to unsupervised hierarchical clustering, a more defined pattern of metabolic alterations induced by exercise was observed. The results revealed significant alterations in nucleotide and amino acid metabolism, glycolysis, and the TCA cycle (Figs. 3C and 3D). A heatmap of all the metabolites is shown in Fig. S1F. In addition, the OCR was significantly lower in shOPA1 than in control cells, and proton leakage and ATP production were decreased (Figs. 3E and 3F). The above results demonstrated that the loss of OPA1 blocked the activity of mitochondria to balance metabolism.

Figure 3 Knockdown of OPA1 decreased the metabolic efficiency in NCI-H1299 cells.

(A) Metabolites were analyzed by PLS-DA. Each principal component is labeled with the corresponding percent values. (B) Variable importance on projection (VIP) represented the importance of the substance in the PLS-DA model. (C) Heatmap analysis of different expression levels of metabolites between Control and ShOPA1 (two-way ANOVA, n = 3). (D) Schematic of glycolysis and TCA cycle process. Green arrows represent down-regulated metabolites. Red arrows represent up-regulated metabolites. (E) Knockdown of OPA1 decreased the aerobic respiration rates as indicated by the OCR. (F) Basal respiration, Proton Leak, ATP production and maximal respiration was lower in OPA1 knockdown NCIH1299 cells. For A–F, n = 3 and the bars represent mean ±SEM. The statistical analysis was carried out using t-test, denotes P < 0.01.

OPA1+ tumor epithelial cells decreased its immune response to CD8+ T cell

To further explore the roles of OPA1 in tumorigenesis, we performed RNA-seq analysis and identified increased activity of immune response pathways (pathway of “immune response” and “MHC class II protein complex binding”) in OPA1 knocked down LUAD cells. GSEA analysis showed the higher activity of immune response and the mRNA expression of related genes, B2M, GBP2, CD24, HLA-F, CTSS, all increased in OPA1 knocked down LUAD cells. Moreover, we used sigle-cell sequencing data to further analysis the immune response of OPA1+ tumor epithelial cell to immune cells. All cells were divided into 19 clusters according to the t-SNE clustering algorithm and exhibited a higher expression of OPA1 in tumor epithelial cell (Figs. 4A–4B). Also, the proportion of OPA1 and MFN1 in various cells tend to be more distributed in tumor epithelial cells. Interestingly, it was found that OPA1 and MFN1 were strongly correlated in tumor epithelial cells (Fig. 4C). We further performed a GO analysis of the significantly different genes in OPA1+ and OPA1− patients in different cell types (Fig. 4D). MKi67 expression is found within proliferating cells alone under general conditions and we found that MKi67 was higher expressed in OPA1+ cells (Fig. 4E). Go analysis of up-regulated and down-regulated genes showed that up-regulated genes mainly played a role in cell adhesion, while down-regulated genes mainly concentrated in immune response-related pathways (Figs. 4F–4G). By comparing the interaction network between OPA1+ and OPA1− tumor tissues, we found that the communication of tumor epithelial with CD8+ T cells was significantly increased in OPA1− tumor tissues (Fig. 4H). This suggests that OPA1− tumor epithelial cells may regulate the immune responses strongly, while OPA1+ tumor epithelial cells are associated with immune escape (Fig. 4I).

Figure 4 OPA1 modulates immune evasion of tumor epithelial to CD8+ T cells.

(A) GO analysis of downregulated gene in OPA1 overexpressed LUAD cells, red represent biological process, green represent cell component, blue represent molecular function. (B) GSEA analysis for immune response pathway. (C) mRNA expression of immune response related gene in control and OPA1 knocked down cells. (D and E) Distribution (D) and proportion (E) of OPA1 gene expression in different cell types; (F) The expression proportion of MKI67 gene in different cell types of OPA1+ and OPA1 −. (G) GO biological process enrichment analysis within the complete set of differentially expressed genes in OPA1 + and OPA1− tumor tissues. (H and I) Capacity for intercellular communication between tumor epithelial cells and other cells. Width of lines indicates the strength of interactions between the indicated cell types.

OPA1 deficiency induced cell death via mitochondrial stress

Having confirmed the decreased activity of mitochondria, we further measured mitochondrial membrane potential. Mitochondrial membrane potential is a key mediator responsible for the activity of mitochondria (Zorov, Juhaszova & Sollott, 2006). As shown in Figs. 5A–5B, using TMRM as a fluorescent indicator for Δψm, we observed a 30% loss of mitochondrial membrane potential Δψm in shOPA1 compared with control cells. With the damaged mitochondrial membrane potential, we detected an increased amount of the cytochrome c protein in the whole cell (Fig. 5C). Cytochrome c is critical in the activation of programmed cell death pathway (Ow et al., 2008a). We further showed that shOPA1 treatment resulted in a significant increase in the percentage of early and late apoptotic cells (about 10.2%) compared to the control group through Annexin V-FITC (Fig. 5F). In addition, we detected the influence on the cell cycle of the damaged mitochondria. Flow cytometric analysis and RT-PCR verified that the cell cycle was blocked in G1 phase (Figs. 5D–5E), accelerating the process of apoptosis.

Figure 5 Knockdown of OPA1 increases apoptosis in NCI-H1299 cells.

(A) Representative confocal images of mitochondria with TMRM staining in Control and ShOPA1 cells . Scale bars, 20 µm. (B) Quantification bar chart of m in Control and ShOPA1 cells. (C) Western blotting detected the expression of and OPA1 and cytochrome c in Control and ShOPA1 cells. GAPDH used as a loading control. (D) Flow cytometry analysis of cell cycle status in Control and ShOPA1 cells. Statistical results are present on right, indicating knockdown-OPA1 cells was significantly stalled at the G1/G0 phase. (E) RT-PCR detected the mRNA expression levels of CDK1, CDC25a and CCND1 in Control and ShOPA1 cells, and GAPDH gene used as a control. (F) Increased apoptosis of NCI-H1299 cells with down-regulated OPA1 in the representative scatter plots of flow cytometry and quantification analysis. For A to E, n = 3 and the bars represent mean ±SEM. The statistical analysis was carried out using t-test, * denotes P < 0.05, ** denotes P < 0.01. m, mitochondrial membrane potential.

OPA1 overexpression shows enhanced proliferation and migration of LUAD cell lines

To evaluate the roles of OPA1 in LUAD, we knocked down OPA1 through virus infection (shRNAs) in the NCI-A549 cell line (Fig. 6A). We found that OPA1 knockdown exhibited significantly reduced cell growth rate (Fig. 6B) and resulted in fewer colonies (Fig. 6C). Moreover, transwell assays showed that knockdown of OPA1 inhibited cell migration compared with the negative control (Fig. 6D). Similarly, inhibition of cell growth, cloning and migration were observed in shOPA1 NCI-H1299 cells (Figs. 6E–6H). In addition, we overexpressed OPA1 in NCI-A549 cells (Fig. 6I), which significantly increased the cell growth rate (Figs. 6J) and colony numbers (6K). Additionally, transwell assays showed that OPA1 accelerated the speed of cell migration compared with the negative control (Fig. 6L).

Figure 6 OPA1 overexpression modulates LUAD proliferation and migration.

(A) mRNA expression levels of OPA1 in Control and ShOPA1 NCI-A549 cells. (B) Cell proliferation was measured by CCK8 assay in Control and ShOPA1 NCI-A549 cells. (C) Colonies were counted in Control and ShOPA1 NCI-A549 cells, statistical results are present on right. (D) Representative images of transwell assays (migration) in Control and ShOPA1 NCI-A549 cells. Statistical results are present on right. Scale bars, 100 µm. (E) mRNA expression levels of OPA1 in Control and ShOPA1 NCI-H1299 cells. (F) Cell proliferation was measured by CCK8 assay in Control and ShOPA1 NCI-H1299 cells. (G) Colonies were counted in Control and ShOPA1 NCI-H1299 cells, statistical results are present on right. (H) Representative images of transwell assays (migration) in Control and ShOPA1 NCI-H1299 cells.(I) mRNA levels of OPA1 after 48 h when transfected with OPA1 overexpression constructs and empty vectors in NCI-A549 cells. (J) Cell proliferation was measured by CCK8 assay in OPA1 overexpression and control NCI-A549 cells. (K) Colonies were counted in OPA1 overexpression and control NCI-A549 cells. (L) Representative images of transwell assays (migration) after 48 h when transfected with OPA1 overexpression constructs and empty vectors in NCI-A549 cells. Scale bars, 500 µm. Statistical results are present on right. Scale bars, 100 µm. For A to H, n = 3 and the bars represent mean ±SEM. The statistical analysis was carried out using t-test, ns denotes no significance, * denotes P < 0.05, ** denotes P < 0.01.

We further tested whether MYLS22, the first in class safe and specific Opa1 inhibitor discovered by Stéphanie Herkenne et al. (Zamberlan et al., 2022), and Oligomycin A, a mitochondrial F0F1-ATPase inhibitor, could recapitulate the effects of genetic OPA1 silencing on breast cancer cell phenotype. A non-toxic dose of MYLS22 inhibited lung cancer cells migration and proliferation (Fig. S3). These results indicate that genetic or pharmacological OPA1 inhibition might reduce lung cancer cell proliferation, migration and invasiveness.

Discussion

Although studies have indicated that OPA1 is a key gene involved in the process of mitochondrial fusion, the loss of which will block the process of metabolism in tumor cells (Li et al., 2020), the underlying mechanism remains poorly defined, and the detailed function of OPA1 in LUAD development remains unknown. In this study, we revealed that knockdown of OPA1 failed to maintain a balance of mitochondrial dynamics and further enhanced the sensitivity of tumor epithelial cells to immune cells.

The morphodynamics of mitochondria, comprising fusion and fission processes, are closely associated with mitochondrial functions (Zong, Rabinowitz & White, 2016). The balance between the processes plays key roles in cancer development (Li et al., 2020). OPA1, located at the IMM, is the key factor for maintaining mitochondrial fusion and preserving cellular health (Wai et al., 2015). It regulates mitochondrial dynamics by governing the delicate balance between fission and fusion of mitochondrial. Li et al. (2020) found OPA1 excessed the activation of mitochondrial fusion in liver cancer and provided a metabolic advantage to sustain tumor growth. An impaired balance caused mitochondrial fragmentation and ultimately resulted in cell apoptosis and were usually observed under stress or pathologic conditions (MacVicar & Langer, 2016). In our study, electron microscopy revealed that the loss of OPA1 caused a reduced mitochondrial fusion and presented a shorter and smaller status, indicating the aberrant metabolism caused by OPA1 deficiency and imbalance of fusion and fission. Pathway analysis also showed that significantly different genes between shOPA1 and control cells involved in mitochondrial function, ATP binding and membrane pathways. These results indicated the effects of OPA1 defects on mitochondria imbalance between fission and fusion and mitochondrial dysfunction. Further, functional assay shows decreased proliferation and migration of OPA1 knocked down LUAD cells which might be induced by the mitochondrial dysfunction and energy unbalance.

The use of CD8+ T cells, with the ability to detect and eradicate cancer cells has been a focus of clinical cancer therapy for over 20 years. Tumors express specific neo antigens1-6 and self-antigens, and CD8+ T cells reactive against such antigens were identified in tumors. In our study, the disturbed mitochondrial dynamics attributed to the loss of OPA1 caused the inhibition of TCA and the mitochondrial respiratory chain, finally significantly decreasing the production of ATP in LUAD. Our present work further reveals that knockdown of OPA1 mediated high expression of immune response related genes contributes to the cell–cell communication of tumor epithelial and T cells.

Metabolic reprogramming fulfills tumors’ energy requirements and is considered a hallmark of cancer (Hanahan & Weinberg, 2011). The activation of several oncogenic signaling pathways, including mTOR, BRAF, and c-Myc has been shown to increase cancer cell glycolysis and lead to lactate accumulation within the tumor microenvironment(TME) (Renner et al., 2017). Recent studies demonstrated that increased lactic acid in the TME could suppress anticancer immune cells by disturbing their intracellular pH (Frauwirth et al., 2002; Gottfried et al., 2006) and impair T cell proliferation and cytokine production (Brand et al., 2016; Fischer et al., 2007). Furthermore, Guo et al. (2022) revealed that tumor immune evasion can be directly regulated by availability of glucose in the TME. When cancer glycolytic activity competes with immune cells for glucose uptake (Husain et al., 2013a; Husain, Seth & Sukhatme, 2013b), cancer cells could effectively achieve immune evasion. In the present study, we identified that the ATP concentration of the OPA1 deficiency group was lower than that of the control group. Further we identified a more activate glycolytic activity in OPA1 overexpressed LUAD cell line. Thus, we speculated that OPA1 overexpression modulates immune evasion is possible due to the increased glycolytic activity. However, more studies were needed to deeply explore the mechanism of how mitochondria dynamics mediated metabolic reprogramming to active immune response of epithelia cells to immune cells. Additional experimental evidence, like co-culture of tumor and immune cells, should be necessary in the further.

In addition to the dysfunction of metabolism, studies have also verified that the loss of OPA1 causes the disorder of fusion in the mitochondrial inner membrane, then cytochrome c release from the cristae and activate the pathway of apoptosis (Kalpage et al., 2020). Cytochrome c is an important proapoptotic protein and also a key component of the OXPHOS system, whose increased release was mediated by inactive of mitochondrial fusion (Ow et al., 2008b; Yadav & Chandra, 2013). However, other studies showed that OPA1 regulates apoptosis by controlling cristae remodeling and cytochrome c redistribution, which is independently from mitochondrial fusion (Frezza et al., 2006). This is in agreement with our results that the amount of cytochrome c increased after knockdown of OPA1. In addition, inhibition of the cell cycle caused by the failure of metabolism jointly led to apoptosis.

In our study, we demonstrated that loss of OPA1 activity results in the dysfunction of mitochondrial morphological and apoptosis, consistent with previously published work (Zamberlan et al., 2022). However, this should not be equated to naturally OPA1 low-expressing tumor cells. Likewise, the inverse argument that higher expression of OPA1 would lead to increased respiration and survival is not necessarily true. Thus, we addressed this limitation and caution against overinterpretation of our work.

Figure 7 OPA1 supports mitochondrial fusion and respiratory function in lung adenocarcinoma.

In tumor cells, the high expression of OPA1 promoted mitochondrial fusion and maintain a healthy mitochondrial network at the inner membrane. The enough production of ATP facilitated the proliferation of tumor cells. In ShOPA1 cells, the loss of OPA1 induced structural and functional abnormalities of the mitochondria and contributed to the apoptosis.

Conclusions

In summary, our results showed that the process of mitochondrial fusion caused by OPA1 was activated in LUAD, which enhanced mitochondrial metabolism to fuel tumor growth and inhibited cell apoptotic pathways (Fig. 7). These findings suggest potential mitochondria-targeted therapy and more effective treatment modalities.

Supplemental Information

Supplemental Information 1 Supplementary Tables

Click here for additional data file.

Supplemental Information 2 Knockdown of OPA1 influences the mitochondrial related pathways and the process of metabolism

(A) RNA-seq data were analyzed by PCA. (B) KEGG pathway enrichment analysis within the complete set of differentially expressed genes. (C) COG_ONTOLOGY pathway enrichment analysis within the complete set of differentially expressed genes. (D) Volcano plot of the metabolites between Control and ShOPA1 cells. (E) Data of the differential metabolites were analyzed by PCA, and different colors of the points represent different sample grouping information, and the arrow direction represents the sample content information of the corresponding substance in the surrounding area. (F) Heatmap for unsupervised clustering of the all the metabolites involved.

Click here for additional data file.

Supplemental Information 3 (A&B) Original immunoblots relative to Fig. 5C

Click here for additional data file.

Supplemental Information 4 MYLS22 and Oligomycin A modulates LUAD proliferation and migration

(A) Cell proliferation was measured by CCK8 assay in Control, MYSL22, and Oligomycin A NCI-H1299 cells. (B) Colonies were counted in Control, MYSL22, and Oligomycin A NCI-H1299, statistical results are present on right. (C) Representative images of transwell assays (migration) in Control, MYSL22, and Oligomycin A NCI-H1299 cells. Statistical results are present on right.

Click here for additional data file.

Additional Information and Declarations

Competing Interests

Author Contributions

Ethics

Data Availability

No potential conflicts of interest relevant to this article were reported. Xiaoxuan Wang participated in the collation of some data and pictures in our study, who is employed by Jiangsu Simcere Diagnostics Co., Ltd., and also declared there are no competing interests.

Ying Wang conceived and designed the experiments, analyzed the data, authored or reviewed drafts of the article, and approved the final draft.

Yadong Li conceived and designed the experiments, prepared figures and/or tables, and approved the final draft.

Xuanwei Jiang performed the experiments, analyzed the data, prepared figures and/or tables, and approved the final draft.

Yayun Gu performed the experiments, authored or reviewed drafts of the article, and approved the final draft.

Hui Zheng performed the experiments, prepared figures and/or tables, and approved the final draft.

Xiaoxuan Wang analyzed the data, prepared figures and/or tables, and approved the final draft.

Haotian Zhang analyzed the data, prepared figures and/or tables, and approved the final draft.

Jixiang Wu conceived and designed the experiments, analyzed the data, prepared figures and/or tables, and approved the final draft.

Yang Cheng conceived and designed the experiments, performed the experiments, analyzed the data, authored or reviewed drafts of the article, and approved the final draft.

The following information was supplied relating to ethical approvals (i.e., approving body and any reference numbers):

The study was conducted according to the guidelines of Geriatric Hospital of Nanjing Medical University.

The following information was supplied regarding data availability:

The RNA sequence is available at NBCI: PRJNA805099.

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
