# Peer review of "OPA1 supports mitochondrial dynamics and immune evasion to CD8+ T cell in lung adenocarcinoma"

_PeerJ, doi:10.7717/peerj.14543_

## Round 0.1 · original submission · Major Revisions

The reviewers provided very good suggestions. These suggestions help to improve the quality of the paper and fully justify the conclusion of the paper. Therefore, this article is suitable for Major Revision.

Reviewer 1 ·

Basic reporting

Overall, the manuscript is well-written with the central messages conveyed clearly. However, grammar checks should be performed such that the writing is up to publication standards. For example, line 32 should state “remain unknown” instead of the singular form of the verb, line 35 should state “single-cell sequencing”, line 36 should state “copy number amplification of OPA1 and MFN1 were co-occurring” or “copy number amplification of OPA1 and MFN1 co-occurred”, etc.

A more complete overview of the relevant topics should be provided in the introduction which in its current state does not sufficiently cover previous findings by those in the field. For example, the authors only briefly described the role of OPA1 in mitochondria fission in their introduction and discussion, and mentioned work by Kalpage et al. in 2020. However, Frezza et al. (Cell 2006 Jul 14;126(1):177-8) reported OPA1’s function in protection against apoptosis via prevention of cytochrome c release, a role independent from mitochondrial fusion. Since the authors’ findings, especially those in Figure 5 and 7, supported this previous report, the authors should provide this background in their introduction and cite the relevant articles including the one by Frezza et al.

Experimental design

The research question is clearly defined, and the authors designed the appropriate experiments to address these questions. The authors might want to reconsider how they frame their research in terms of how it fills knowledge gaps once they provide a more complete overview of the existing literature (detailed comments in the Basic Reporting section).

The method section lacks sufficient details. The sequences for the OPA1 shRNAs used in the knockdown experiments should be provided. The analysis used for assessing capacity for intercellular communications in Figure 4 H and I was not included in the methods section.

Validity of the findings

The experiments are generally well-designed and well-executed.

The contrast between OPA1 knockdown cells and OPA1 high-expressing tumor cells should be made with more care. The author’s data along with previously published work demonstrated that loss of OPA1 had a clear impact on cell metabolism and survival. However, this should not be equated to naturally OPA1 low-expressing tumor cells. Furthermore, the inverse argument (that higher expression of OPA1 would lead to increased respiration and survival) is not necessarily true and can only be addressed by overexpression, not loss of function, experiments. Therefore, the authors should state the limitations of their study and caution against overinterpretation of their results.

Reviewer 2 ·

Basic reporting

The overall outline and flow of the content are natural.

Experimental design

Research questions are well-defined, relevant and meaningful. However, it is believed that if there was a study that inhibited OPA using chemical, it could have been close to the mitochondrial target treatment that researchers wanted to reveal.

Validity of the findings

Conclusions are well stated, linked to original research question & limited to supporting results.

Additional comments

This study well expresses the importance and association of mitochondria through various research methods. The development or flow of the content is clear, but it is difficult to find novel thing. However, I think the purpose of the study is to prove their specificity in that it is a study tor mitochondrial target treatment.

Reviewer 3 ·

Basic reporting

Authors of the manuscript titled “OPA1 supports mitochondrial dynamics and immune evasion to CD8+ T cell in lung adenocarcinoma” address an important topic on health, the apoptosis resistance of tumor cells, but in relation to the mitochondrial dynamics. Although manuscript is well writing, there are some mistakes here and there indicating that a carefully review is necessary, i.e., lines 124-125, is write as “…cells in different groups were seeded in 75 culture dishes and treated with different concentrations of NCI-H1299 (0, 2.5, 5, and 10uM)”; however . NCI-H1299 are cells, not a treatment. On lines 128-129, “After resuspending in PBS and counting, 2×105 129 cells were centrifuged and supernatant.”… seems like the paragraph is unconcluded.
The results are clear the most, and figures are well organized; unfortunately, some of the text in sections are not clearly legible probably because of the font size, i.e., Figure 4.
There is not possible to identify a research question on the Introduction section, but there is a reference to this important part of the research in the abstract, “…the genomic mechanism of its activation and the roles in immune evasion in non-small cell lung cancer remains unknown.”

Experimental design

Experimental design is adequate for the goal of the study, but there are observations to clarify in order to improve the manuscript. Mitochondrial dynamics is a dual phenomenon integrated by fusion and fission, in the Figure 1A, expression of OPA and MFN is increased in tumor, but also DNM1L (Drp1) is overexpressed; however this finding was ignored by authors and, even more, there is another important fission-related protein Fis1 that was neither analyzed nor mentioned, please explain.
Figure 1 title is “The high expression and function of OPA1 in LUAD.” But there is no one experiment to demonstrate any functionality of OPA
Images in Figure 2F does not represent the graphical data, also the bar introduced for indicate the scale is impossible to see. Must consider including pictures that are more representative.
Authors indicates in Figure 4 that OPA expression modulates immune evasion, but it is possible that mitochondrial dysfunction, induced by any other mechanism has the same result, in that circumstances, additional experimental evidence should be necessary, please discuss.
In Figure 5A and B, authors shown that OPA silencing induces almost complete mitochondrial potential collapse, however that does not match completely with the OCR in Figure 3E, where oxygen consumption decreases around 20-30%. In addition, picture of ShOPA1 is not representative of the Relative TMRM fluorescence.
On the other hand, western blot must be analyzed carefully because OPA1 is a 80-100kDa protein, whereas in the figures, it is pointed in 37kDa, almost the same size of GAPDH. Please correct or explain this discrepancy.
The measurement of Cyt C is not determinant as apoptosis marker, instead change in its intracellular location (from mitochondria to cytosol) must be demonstrated.
Figure 6 indicates “OPA1 modulates LUAD proliferation and migration.”, however that may be a miss- or over-interpretation because the OPA silencing induces mitochondrial dysfunction, and energy unbalance, which finally trigger apoptosis and reduces cell grown and migration.
Figure 7 does not indicates the effect of OPA over regulation on fission and fusion, but only the effect on mitochondrial energy metabolism; the row corresponding to fusion should be bigger than the fission one.
The Discussion section is so limited and must be extended in order to cover the possible imbalance between fission and fusion, the effect of mitochondrial dysfunction on the analyzed variables, because mitochondrial dysfunction is the main result of an increased fission (or OPA down regulation) and is difficult to propose a direct effect of changes in OPA expression.

Validity of the findings

3. Validity of the Findings
The results of this study are relevant because of the field of research, although similar results were published before (Li M, Wang L, et al. Mitochondrial Fusion Via OPA1 and MFN1 Supports Liver Tumor Cell Metabolism and Growth. Cells. 2020 Jan 4;9(1):121, ., the relevance in still significant, but interpretation of results must be improved, among the discussion section.
In conclusions (line 308) authors claim the demonstration accelerated cyt c releas from mitochondria, however, experimental approach just demonstrate an increased amount of the protein in the whole cell because any cellular fractioning was developed. In addition, in the discussion section, line 292-293 indicates “Our electron microscopy results indicated that the loss of OPA1 resulted in a disordered number and morphology of mitochondria. Mitochondria showed a decrease in number and fragmentation” which is a miss-interpretation of the results because neither quantification nor morphological study was developed and, as mentioned before, the single electron microscopy picture showed no not support that conclusion.

Additional comments

None

---

## Round 0.2 · accepted · Accept

Thanks for the careful revision.

Reviewer 1 ·

Basic reporting

The authors have addressed my previous concerns regarding grammatical errors and incomplete introduction.

Experimental design

The revised manuscript contained an improved description of the scope of their research and more detailed descriptions of their research methods.

Validity of the findings

My previous concerns regarding data interpretation have been addressed.

Reviewer 2 ·

Basic reporting

no comment

Experimental design

no comment

Validity of the findings

The author reinforced sufficient supplementary materials and explanations for each reviewer's questions and requirements. Therefore, it is judged to be suitable for publishing in this journal.

Additional comments

no comments.

Reviewer 3 ·

Basic reporting

No comment

Experimental design

Previous recomendations were solvented.

Validity of the findings

No comment

Additional comments

All previous observations were well corrected or solved.